# Segmentation of Dental Restorations on Panoramic Radiographs Using Deep Learning

**DOI:** 10.3390/diagnostics12061316

**Published:** 2022-05-25

**Authors:** Csaba Rohrer, Joachim Krois, Jay Patel, Hendrik Meyer-Lueckel, Jonas Almeida Rodrigues, Falk Schwendicke

**Affiliations:** 1Oral Diagnostics, Digital Health and Health Services Research, Charité–Universitätsmedizin Berlin, 10117 Berlin, Germany; csaba.rohrer@charite.de (C.R.); joachim.krois@charite.de (J.K.); jorodrigues@ufrgs.br (J.A.R.); 2ITU/WHO Focus Group on AI for Health, Topic Group Dental Diagnostics and Digital Dentistry, 1202 Geneva, Switzerland; 3Informatics, Department of Health Services Administrations and Policy, Temple University College of Public Health, Philadelphia, PA 19140, USA; patel.jay@temple.edu; 4Operative, Preventive and Pediatric Dentistry, 3010 Bern, Switzerland; hendrik.meyer-lueckel@zmk.unibe.ch; 5Surgery and Orthopedics, UFRGS, Porto Alegre 90040-060, Brazil

**Keywords:** machine learning, deep learning, image segmentation, dental restorations

## Abstract

Convolutional Neural Networks (CNNs) such as U-Net have been widely used for medical image segmentation. Dental restorations are prominent features of dental radiographs. Applying U-Net on the panoramic image is challenging, as the shape, size and frequency of different restoration types vary. We hypothesized that models trained on smaller, equally spaced rectangular image crops (tiles) of the panoramic would outperform models trained on the full image. A total of 1781 panoramic radiographs were annotated pixelwise for fillings, crowns, and root canal fillings by dental experts. We used different numbers of tiles for our experiments. Five-times-repeated three-fold cross-validation was used for model evaluation. Training with more tiles improved model performance and accelerated convergence. The F1-score for the full panoramic image was 0.7, compared to 0.83, 0.92 and 0.95 for 6, 10 and 20 tiles, respectively. For root canals fillings, which are small, cone-shaped features that appear less frequently on the radiographs, the performance improvement was even higher (+294%). Training on tiles and pooling the results thereafter improved pixelwise classification performance and reduced the time to model convergence for segmenting dental restorations. Segmentation of panoramic radiographs is biased towards more frequent and extended classes. Tiling may help to overcome this bias and increase accuracy.

## 1. Introduction

A wide range of applications in the field of artificial intelligence and, specifically, machine and deep learning (ML/DL) are entering health care. A particularly prolific subfield of ML/DL is computer vision, i.e., image analysis using ML/DL methods. Computer vision applications have been developed in dermatology (for analyzing skin photographs), ophthalmology (retinal imaging) and oncology [1,2,3]. In dental radiology, ML/DL has been increasingly adopted to analyze dental radiographs—for instance, to detect caries, apical lesions, periodontal bone loss or anatomic landmarks on a wide range of 2D and 3D imagery, including panoramic radiographs [4].

Image segmentation refers to classification tasks for pixels with semantic labels (semantic segmentation) or partitioning of individual objects (instance segmentation) [5]. Semantic segmentation performs pixel-level labeling with a set of object categories for all image pixels (e.g., human, car, tree, sky, and, as in the present case, different dental restorations). Instance segmentation extends the semantic segmentation scope further by detecting and delineating each individual object of interest in the image (e.g., partitioning of each individual dental restorations). For medical applications, image segmentation is popular given that the model output (segmentation masks) can be interpreted by human experts and hence provides inherent explainabilty. The downsides of such an approach are (1) the high human expert labor costs, as each object of interest needs to be annotated by marking the area; (2) classical evaluation metrics fall short as the pixel scale as interpretation is much more difficult, hence concepts and metrics such as the intersection over union (IoU) and (weighted) mean average precision (mAP) are applied [6]; and (3) technical challenges emerge, as only a few pixels of the image actually show the area/lesion of interest in many use cases, which in turn results in an imbalanced learning task and hence a more challenging task, where many negative pixels but only a few positives need to be learnt.

While assisting practitioners in pathology detection or anatomic landmarking has obvious diagnostic value, ML/DL computer vision applications also come with the promise of increasing the efficiency and quality of reporting. To support dentists in systematic reporting of radiographic findings, a range of further ML/DL models are required—for example, for tooth detection and classification or the detection and classification of restorative materials and previous dental work experience. Having such models allows one to assign identified pathologies to teeth and to comprehensively report all findings from the radiograph in only a few seconds, which is relevant for more extended imagery such as panoramics. An increasing number of studies reported on the detection of pathologies, while fewer studies are available for detecting and classifying dental restorative materials for example [7,8].

Current ML/DL models for computer vision mainly employ Convolutional Neural Networks (CNNs). CNNs have a fixed resolution for the input image, which is smaller than the original image in most cases (especially for large imagery such as panoramics). Hence, images must be downsized before applying the CNN. It is conceivable that cropping larger images (such as panoramic radiographs) into smaller, equally spaced rectangular image crops (tiles) and training CNNs on such crops circumvents information loss due to downsizing and hence improves model performance.

Using the exemplary task of semantic segmentation of restorations and other previous dental work (such as fillings, crowns, root-canal fillings and implants) on panoramics, we hypothesized that models trained on tiled data outperform models trained on full radiographs.

## 2. Materials and Methods

### 2.1. Study Design

The goal of this study was to improve the accuracy of multi-label segmentation of dental restorations on panoramic radiographs by using a tiling approach. The original radiographs were cropped into multiple parts to increase the resolution of the images used during the training of the model. By using these subsets of the original images, we aimed to capture more details, thus increasing segmentation performance. We trained on a different number of tiles between 2 and 20 and evaluated our models using the full image. The experimental workflow is shown in Figure 1. Reporting of our study follows the respective checklists [9,10].

### 2.2. Dataset

Our dataset contained 1781 panoramic radiographic images. Data collection was conducted between 2015 and 2018 and was ethically approved (ethics committee of Charité Berlin, EA4/080/18). Panoramics involving at least one permanent tooth were included. There were 54.3% male and 45.7% female patients; the mean (SD, min–max) age was 54.3 (19.7, 10–95) years. Data were generated by machines from the manufacturer Dentsply Sirona (Bensheim, Germany).

The images were annotated pixelwise by dental experts using an in-house custom-built annotation tool [11] for four different restoration types, namely fillings, crowns, root canal treatments, and implants. These four annotations were stacked together to create multi-channel segmentation masks. Each of the channels represents the detections, or lack thereof, with pixelwise true and false values. The channel values are not mutually exclusive, so a pixel can belong to multiple classes in case of overlapping annotations. This means we were solving a multi-label semantic segmentation problem.

### 2.3. Model

The selected model for our experiments was based on U-Net [12], a Convolutional Neural Network for semantic image segmentation. It consists of an encoder and a decoder part connected with skip connections. The encoder extracts features from different spatial resolutions which are used by the decoder to define an accurate segmentation mask. The encoder is also known as the backbone since it is usually pretrained on a sizable dataset for feature extraction. For our backbone, we used the ResNeXt-50 (32 × 4d), a variation of the ResNet-50 architecture with an added cardinality dimension [13]. The applied backbone was based on the pretrained implementation of ResNext on the ImageNet dataset [14] from the PyTorch deep learning framework.

### 2.4. Training

We performed a five-times-repeated three-fold cross-validation (i.e., three-fold validation was repeated 5 times) to obtain more robust performance metric estimates and to reduce the bias introduced with a single training, validation, and test set partitioning. First, images were split into five groups. Each group represented a hold-out test of 20% of the dataset. We repeated a three-fold cross-validation for each test set, where the remaining four groups formed the training-validation data. These data were further split into actual training and validation images in three different splits. Each repetition resulted in three trained models, which were evaluated on their respective hold-out test set (Figure 2). Since we employed this method on all the test groups, our approach yielded fifteen models, which we used to establish mean and 95% confidence level of reported performance metrics.

We trained the decoder and fine-tuned the encoder side of our network. Before the panoramic radiograph images and their masks were fed to the model as input, we created smaller, equally sized crops (tiles) from them. We used an incremental number of tiles in our experiments from 2 up to 20. We also trained on the full images without any cropping for comparison, hence we had 11 different tile settings. The described cross-validation scheme was used for each of these settings.

The tiles would become the input to the network. The input images were resized to the resolution of 224 × 224 and then preprocessing functions (input range selection, normalization) were applied. The Dice loss function was used for training. The parameters were iteratively optimized using the Adam optimizer, with the initial learning rate and momentum set to 10 × 10^−4^ and 0.9, respectively. The batch size used was 50. We used an adaptive learning rate based on the calculated validation loss after each epoch. If there was no improvement in the last five epochs, the learning rate was reduced to 10 × 10^−5^. We employed the same strategy for early stopping as well. Once the learning rate was reduced, after no improvement in the last five epochs, we stopped the training process. This allowed us to keep optimizing the parameters further and save time by terminating the training loop. To increase the diversity of the data and to help prevent overfitting, we used data augmentation techniques during the training phase. The augmentations were applied to the original training data randomly to generate more examples. The techniques used were affine transformations, e.g., horizontal flips, rotation, brightness/contrast manipulations, image blurring and sharpening and gaussian noise application. The setup of the model architecture and optimization process was carried out using the deep learning framework PyTorch (1.4.0) and the Python (3.7.3) programming language. The models were trained using Nvidia A100 graphics cards with CUDA 11.

### 2.5. Metrics

We employed different metrics to compute the performance of our model. We used class-wise and overall F1-score, i.e., the harmonic mean of positive predictive value (precision) and sensitivity (recall), as well as sensitivity and, on a pixel level, intersection over union (IoU). The metric scores were evaluated on the different mask channels individually, and together as well.

### 2.6. Evaluation

For the evaluation, we used our trained model for inference on our test set. We cropped the images with the appropriate number of tiles, computed the segmentation masks, and calculated our evaluation metrics. Since we are dealing with a semantic segmentation task, we did not consider individual restorations separately but the whole class segmentation instead. We calculated pixelwise metrics on the image compared with the different channels of our predicted segmentation mask.

### 2.7. Explainability

A frequent problem with deep learning models is that the underlying neural networks are black boxes, meaning they do not provide insights into the approximation given. This is especially noticeable on classification problems, where the output of the model is a yes or no, or a simple label. In the case of segmentation, where the output is an area on the input image, the transparency of the prediction is an inherent property. For this reason, no further explainable techniques were applied to our models.

## 3. Results

Our experiments showed that with increasing number of tiles, model performance increased significantly. Using only one tile, the mean F1-score over all channels was 0.7 compared to 0.95 when 20 tiles were used. The most prominent increase occurred for root-canal fillings, i.e., small cone-shaped, less frequent features. In their case, the mean F1-score increased from 0.33 to 0.97. For implants and crowns, no significant improvement was detected. Sensitivity also showed a similar increase from 0.64 to 0.95 (Figure 3). 

IoU on all channels also increased from 0.59 to 0.93 when using one to 20 tiles. All metrics for different restoration types are presented in the Appendix A.

While the number of input images per training epoch increases with higher number of tiles, we found tiling to nevertheless reduce convergence time, i.e., increase training efficiency (Appendix A). 

## 4. Discussion

For many medical ML/DL applications involving imagery (computer vision), downscaling image size and resolution, i.e., information loss, is required for training. This is also true for larger dental radiographs such as panoramics radiographs. Such loss of information may be well accepted for detection or segmentation tasks of larger and more prevalent entities (such as larger bone structures or, in dental imagery, teeth, crowns or bridges). For smaller and less prevalent findings such as root-canal fillings, but possibly also caries or apical lesions, this may be detrimental to the learning outcome, i.e., the diagnostic performance of the developed ML/DL application. Semantic segmentation of dental restorations and other previous dental work is a clinically relevant problem, as it allows to generate an automated and comprehensive. We hypothesized that downscaling imagery may negatively affect the segmentation accuracy of ML/DL for this task, and that tiling panoramic images may overcome this problem. 

We confirmed this hypothesis; segmentation accuracy was significantly higher when considering all restoration classes (channels) for models trained on tiled than non-tiled images. A higher number of tiles showed a positive association with accuracy. Notably, when assessing the different classes, this effect was more prominent for smaller objects such as root-canal fillings or fillings than larger ones such as crowns or implants, as expected. Moreover, performance increased for both sensitivity and F1-score. Pixel-wise metrics such as the IoU also reflect that behavior. Notably, specificity and relating metrics were not estimated, as in semantic segmentation the negative class is made of the background. We further found tiling to speed up training convergence, a possible measure of efficiency. 

Our findings need more detailed discussion. First, it is conceivable that the beneficial effect of tiling may be transferable to other segmentation tasks in the dental realm with class imbalances (a large number of pixels representing the background while only a few pixel represent the target), e. g., apical and caries lesions.

Second, our tiling approach reduced model convergence time, as using more tiles reduced the number of epochs needed for training.

Third, it needs highlighting that our study did not aim to develop clinically applicable segmentation models (maximizing accuracy), but that we rather aimed to systematically assess the effect of tiling to retain as much image information as possible by circumventing the need to downscale the image. Future developers interested in clinical applicability may employ our approach and build on larger and more diverse datasets, but also leverage further instruments (such as hyperparameter tuning) to increase accuracy.

This study comes with a number of strengths and limitations. First, it was the first study to assess the effect of information retention (via tiling) for a widely employed semantic segmentation ML/DL task in dentistry. Second, and as discussed, the focus of this study was experimental, and our controlled approach with restricted margin conditions was not aimed for developing clinically applicable models. Last, and as a weakness, we employed data from one center and one x-ray manufacturer. Generalizability to other datasets or ML/DL tasks may not be given.

## 5. Conclusions

In this study, we explored the application of a U-Net architecture in a semantic segmentation setting to a varying number of image crops of panoramic dental radiographs. By tiling the original image and its corresponding segmentation mask, then cropping it for input to our neural network, we effectively increased the resolution. Our results showed that with a higher number of tiles, the model’s performance improved. Smaller and less frequent features that might go unnoticed on the full panoramic image were especially affected. This approach could be easily tailored to other applications to raise detection performance.

## Figures and Tables

**Figure 1 diagnostics-12-01316-f001:**
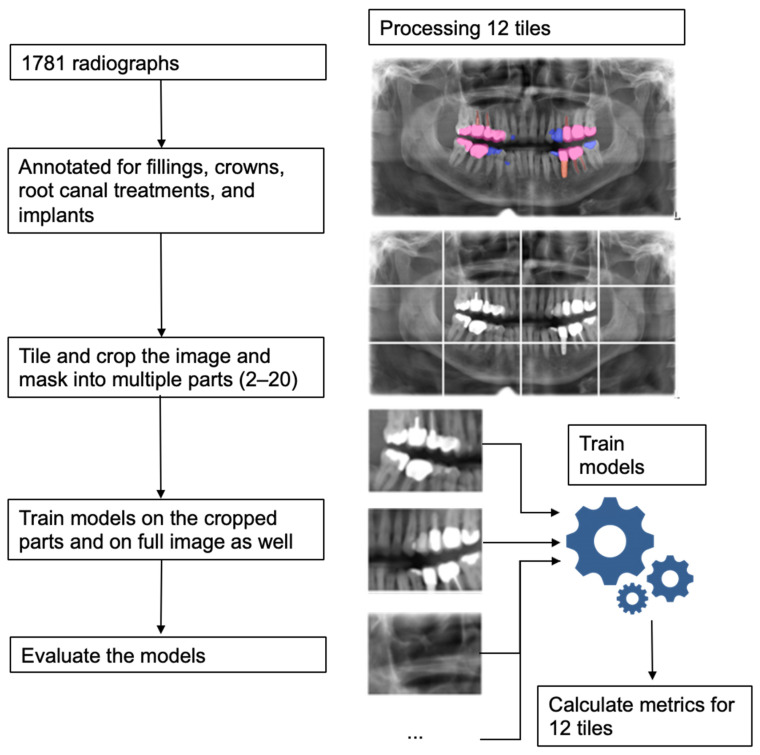
Experimental workflow and an example showing processing an image with 12 tiles.

**Figure 2 diagnostics-12-01316-f002:**
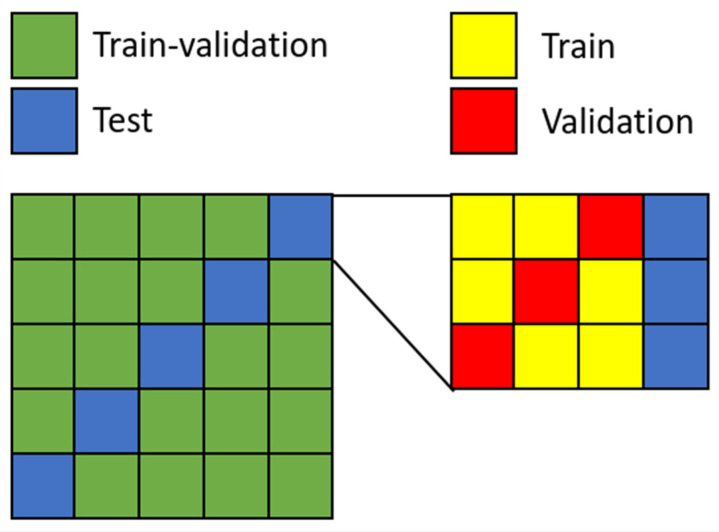
Visualization of the five-times-repeated three-fold cross-validation. For each hold-out test set, the remaining data is split into three different train and validation sets. Each datapoint is used as test data at some point.

**Figure 3 diagnostics-12-01316-f003:**
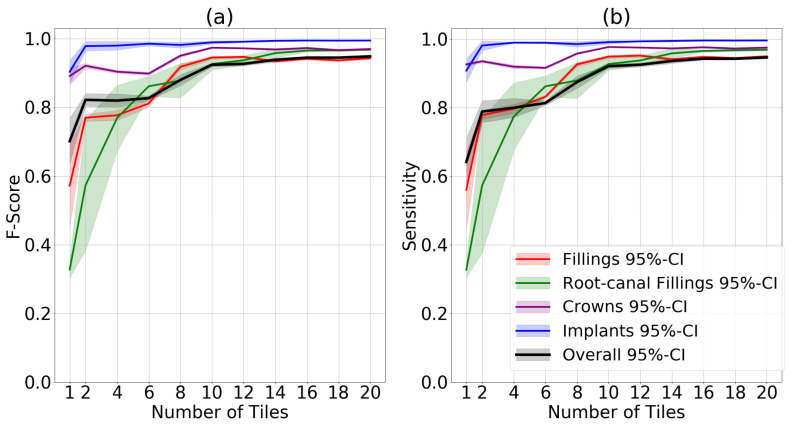
Mean and 95% CI F1-score (**a**) and sensitivity (**b**) for each restoration class as well as for all classes jointly.

## Data Availability

The data are not publicly available due to data protection reasons.

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
