# Peer review of "Segmentation of Dental Restorations on Panoramic Radiographs Using Deep Learning"

_diagnostics, 2022, doi:10.3390/diagnostics12061316_

Round 1
Reviewer 1 Report
the manuscript requires extensive language editing
provide more context about semantic segmentation and its advantage & disadvantage
in the introduction correct the spelling of anatomic landmarking
why it is mentioned as 5times 3 fold cross-validation ? is it not 5 fold cross-validation? provide context

Author Response
Comment: The manuscript requires extensive language editing.
Our response: We revised the manuscript.
Comment: Provide more context about semantic segmentation and its advantage & disadvantage.
Our response: This was expanded on.
Comment: In the introduction correct the spelling of anatomic landmarking .
Our response: Done.
Comment: Why it is mentioned as 5times 3 fold cross-validation ? is it not 5 fold cross-validation? provide context
Our response: This was revised.
Reviewer 2 Report
This study is a customized object segmentation study for dental diagnosis support, and I agree that it will be accepted in this journal without any additional information in its current state.
Author Response
Comment: This study is a customized object segmentation study for dental diagnosis support, and I agree that it will be accepted in this journal without any additional information in its current state.
Our response: Many thanks.